# Integrated mental health for refugees: A realist theory building study

Nancy Clark[1]*, Alejandro Argüelles Bullón[2], Mita Huq[3], Ferdinand C. Mukumbang[4]

1 Department of Nursing, Faculty of Health, University of Victoria, Victoria, British Columbia, Canada,
2 Division of Health Research, Faculty of Health and Medicine, Lancaster University, Lancaster,
United Kingdom, 3 Department of Global Health, University College London, London, United Kingdom,
4 Department of Global Health, University of Washington, Seattle, Washington, United States of America

* nancyclark@uvic.ca

## Abstract

People with forced migration backgrounds, such as refugees, experience disproportionate mental health conditions related to complexities associated with acculturation, separation from family, traumatic events due to war or persecution and precarious journeys in their effort to find protection and care. Intersecting social determinants of refugee mental health include navigating and finding health care resources, employment, housing and social support. Because of the complexity of health and social needs that refuges experience, there is a need for robust integration of mental health services across services such as settlement organizations and primary health care services. Robust service integration to address mental health for refugees can benefit from a theory-driven approach to understanding integrated mental health service delivery. This study engaged in deliberative dialogues with multidisciplinary interest group holders from settlement services, primary health care, mental health, a survivor advocacy group and a policy analyst (N = 24) to understand how services work to promote refugee mental health in a Canadian context. Adopting a participatory realist approach, we developed an initial program theory on the integration of refugee mental health across services. We found trust, connection, proactivity and moral commitment to be key mechanisms that enabled better integrated mental health care across refugee clients, providers and services. Mechanisms which hindered integration included alienation, stagnation, burnout and fragmentation. Findings indicate that, when funding is allocated to settlement programs, supports like cross-cultural brokers, community health workers and navigators can then be implemented. These resources then address social determinants of refugee mental health and trigger positive mechanisms for equitable, just policy approaches to integrate services for refugee mental health.

**Data availability statement:** All relevant data are within the manuscript and its Supporting information files.

**Funding:** The author NC received funding fo rthis work through the Michael Smith Health Research BC Convening and Collaborating C2 Award Grant #C2-2022-2929. The funders had no role in the study design, data collection and analysis, decision to publish, or preparation of the manuscript.

**Competing interests:** The authors have declared that no competing interests exist.

## Introduction

Currently, approximately 123.2 million people need international protection due to war, human rights violations, climate emergencies and political persecution [1], leading to significant mental health conditions across all contexts of forced migration [2–4]. The International Organization for Migration (IOM) defines refugees and asylum seekers as forced migrants because they have been forced to move due to disasters, human rights violations, and violence and therefore require legal protection [5]. Both categories of forced migrants encounter traumatic experiences throughout their migration journey including resettlement contexts which can increase risk of developing mental health challenges [6]. Approximately one in five people living in settings affected by conflict experience poor mental health [1]. A recent systematic review found an estimated prevalence of Post-Traumatic Stress Disorder (PTSD) and depression for refugees and asylum seekers greater than general population: 31% for PTSD and 31.5% for depression [7]. Mental health stigma remains one of the most cited reasons why refugees fail to access mental health services, however, cultural stigma may overshadow social and structural conditions such as political and historical factors that reinforce stigmatizing attitudes [8,9]. Studies suggest that health care systems have not been responsive in addressing specific health needs of forced migrants, including lack of universal health care, interpreters, gender or culturally safe, trauma informed practices [10–13]. Thus, every host country's health system requires deliberate planning and adaptation to provide mental health services for migrant populations [14]. Such deliberate planning and organization can address primary care providers' and settlement workers' challenges in providing mental health support to refugees related to lack of time, lack of knowledge about universal health coverage, lack of gender responsive services, limited integration of language resources and interpreters, and limited understanding of the political and cultural background of the refugee group [3]. The lack of integration of care across systems and services can address barriers to care but require stable funding and support to ultimately decrease healthcare costs and better mental health for forced migrants [13]. In some settings, refugees, asylum seekers and other people experiencing forced migration are denied care based on their 'legal' status [14]. Access to mental health services is a human right [3,4,6], therefore, it is necessary to improve existing mental health supports and services for refugees.

### Refugee mental health care services

De-prioritizing social determinants of refugee health can decrease alignment between interventions and causes of poor mental health outcomes for culturally diverse migrant groups [15]. Mental health service research tends to focus on white cisgender, heterosexual, middle-class norms, thus overlooking how mental health needs differ across other intersections of identity such as those of forced migration backgrounds and minoritized cultural and ethnic identities. There exists a gap in knowledge that addresses health system and services programming to promote refugee mental health, which can lead to further marginalization and decreased access to care [16,17]. Importantly, there also exists a gap in theory-informed models of mental health services for refugees.

Silos between and within community health programs, including primary health care services, can also lead to system and service barriers related to fragmentation. Better integrated mental health care may facilitate increased collaboration and coordination of care for refugee population groups. Mental health integration must include universal health coverage and equitable access to services, along with tailored supports that meet the intersecting social determinants of refugee mental health such as cultural stigma, financial insecurity and language barriers [4,16–18].

In line with the priorities and objectives of the World Health Organization Global Action Plan 'Promoting the health of refugees and migrants' (2019–2023) and the WHO 'World Mental Health Report: Transforming health for all,' there is a need to improve coordination across sectors beyond the health sector to promote and provide integrated mental health care for refugee and asylum seekers [19–21]. Coordination of mental health services should include mainstreaming refugee health and system level capacity to address social determinants of health [20,21]. The complexity of refugee needs in post-migration contexts requires an integrated approach to improve accessibility, reduce system fragmentation, avoid duplication of resources and provide overall equitable health care resources for refugees. Equitable access to health care services is also considered a structural determinant for refugee health [21]. Canada has historically been a leader in refugee resettlement, with integration as a key immigration policy objective [22]. Despite initiatives to provide specialized refugee health clinics and expanded settlement services, recent cuts in government funding to the settlement and immigration sector may constitute a crisis that limits resources.

## Background and literature review

**Integrated care.** Theories underpinning integrated care suggest that better integration of services can promote quality care, reduce fragmentation, and improve both health outcomes and cost efficiency [23,24]. The concept of integrated care emerged from the primary health care movement to provide accessible contact, comprehensiveness, continuity and service coordination [23]. There is no universal definition of integrated care. It can include 1) increased coordination across intersecting systems; 2) patient family networks; and 3) inter-organizational referral networks as models that decrease service fragmentation, improve coordination, and provide greater effectiveness and improved patient care [20,24].

Integrated health care can be implemented in many ways and at various levels of a health system [25]. For service users, integrated care can include a person-centered approach and whole person care that are coordinated across or within settings. For health care organizations, integrated care is about responsibilities: multidisciplinary teams, task sharing, links to social networks, common information systems and professional partnerships based on shared roles, for example. For ministries of health, integrated care is about having joint policies, financing mechanisms and governance structures across physical and mental health services [20].

Some studies on integrated care focus in primary care contexts suggesting that collaborative approaches promote better engagement with available services and continuity of care [26–28]. Studies also highlight the need for structured settlement support and programs beyond primary care, including intersectoral collaboration to critically address the social, political, economic and cultural determinants shaping forced migrant mental health [16,26]. Increasingly, realist approaches have been used in health services research to investigate health system fragmentation and the complexity of interrelations between health system actors and policy [29,30]. Before describing the known mechanisms or determinants of integrated care, we briefly discuss realist theory.

A realist approach recognizes the micro (individual), meso (organizational) and macro (system) contexts that trigger mechanisms (reasoning applied to resources, opportunities and constraints) of various actors to implement change [31]. The mechanisms, i.e., reasoning and resources, opportunities and constraints are key components of a program architecture that structure a given intervention [32,33]. Realist theory refers to mechanisms, understood as the combination of reasoning and resources, opportunities and constraints by people that trigger change behaviors. However, context that promote change are the social, cultural, institutional or relational dynamic conditions that shape whether how and

for whom those mechanisms (reasoning, resources) generate outcomes and are shaped by practitioners and actors who implement health programming [34,35].

A central premise to realist theory building is the idea of generative causation, that is, the forces which cause something to happen in a particular context, are not readily observable [36]. Thus, realist theory can be used to unearth the mechanisms that lead to systems change for integrated mental health services. Importantly, developing a program theory can be a useful framework that describe what a program or policy is and how it is expected to work [37]. Realist theorists advocate for discerning and explaining the specific contexts that create or enable change so that favorable health outcomes can be implemented [33,34,38]. Understanding how integrated mental health services work for refugees can help to address the disproportionate barriers to access and uptake of mental health services in resettlement settings. Our theory building approach is guided by realist terminology (Table 1).

**Known mechanisms related to integrated care.** Mental health is a top research priority amongst refugees across all phases of resettlement [22]. Recommendations from service users include system navigation and resources with better connection to primary care clinics, insurance coverage, addressing human resource shortage of physicians, embedded professional translation services, and development of refugee advisory boards [14]. Although these findings echo previous health systems and services research, not all contexts and settings are the same. What is effective in one setting may, in another, may not be the same mechanisms that promote integrated mental health care for refugees. The distribution and coordination of mental health care services can vary between and within in high and low-income countries and are dependent upon governmental funding arrangements and policies that explicitly address health care inequities fort vulnerable population groups.

Studies have shown that crisis contexts, such as COVID-19, have had a ripple effect on mental health service delivery, contributing to a change in the way services are delivered to address heath and health care inequities for marginalized population groups [43–45]. A realist review of the literature during COVID-19 found that trust, social connectedness, accountability and resilience were key mechanisms that promoted mental health across policy (macro-level), community (meso-level) and individual (micro-level) settings [45]. The same research suggests that pre-COVID-19 provided

**Table 1. Realist definitions[a].**

| Term | Definition |
| --- | --- |
| Context | The social, cultural, institutional, relational and material conditions, both observable and dynamic, that interact with mechanisms to influence whether, how and for whom an intervention works. |
| Mechanism | For this project, a mechanism is the combination of resources, opportunities and constraints introduced by an intervention and the reasoning or response of individuals to them, which, when activated in the right context, generates outcomes. |
| Outcome | The observed effects or changes, intended or unintended, that result from the interaction between mechanisms and context. |
| ICAMO statement | A heuristic tool that helps the realist researcher to theorise the intervention of interest, the context in which the mechanisms may be operating, the relevant actors through whom the intervention is expected to work or is implemented, the mechanisms that are likely to fire and the outcomes (intended or unintended). |
| Programme theory | An explanation of how and why an intervention or programme is expected to work, typically structured using a Context–Mechanism–Outcome (CMO) configuration or a similar heuristic. |
| Retroduction | An explanatory inference in which the researcher goes beyond surface-level observations to identify and explain the underlying, often hidden, forces or mechanisms that may have interacted to produce the observed outcomes. |

[a]Definitions from Dalkin et al. 2015 [39]; Mukumbang, 2019 [35]; Coleman et al. 2020 [40] Greenhalgh et al. 2022 [41]; Mukumbang, 2023 [42].

a context that stimulated different mechanisms, such as power sharing among government and service organizations ultimately leading to the mobilization of resources for vulnerable population groups experiencing chronic care conditions and mental health challenges [45]. This research further suggests that the mechanisms of power-sharing and relationship-building were absent during COVID-19 due to competing accountability factors. These factors were related to the macro (policy/government) level and its need to provide crisis management. However, program outcomes related to resilience and responsiveness were connected to relational reserves amongst interest holders, causing better system adaptation and collaboration with respect to mental health interventions [45]. Across all levels, trust and social connectedness were mechanisms that promoted mental health and were dependent upon pre-existing relationships between multidisciplinary providers and services users [45]. Similarly, trust has been found as a mechanism for external and internal interactions amongst vulnerable ethnic minority groups ability to use mental health services in both middle and low-income countries [31].

Findings from a recent narrative review suggest that system fragmentation between health and social care services could be addressed through better data-sharing approaches, better integration of governance and partnerships; better integration of budgets that cover targeted population groups and an integrated workforce with expanded roles [46]. Another realist review showed that compassion was a key mechanism that stimulated collaboration between multidisciplinary organizations in the context of mental health crises [47]. Other research on mental health interventions with refugees, also showed that community-based mental health services for refugees work when the following are in place: 1) the provision of migrant-sensitive health care services, i.e., language interpretation and culturally tailored support; 2) task shifting and the inclusion in intervention delivery of non-medical personnel such as social workers, nurses and people with lived experience; 3) cultural adaptations that meet refugee community needs; and 4) increased intensity of psychotherapy interventions. Conversely, mechanisms that hindered service impact included 1) lack of gender responsive interventions; 2) lack of mentorship, advocacy and support—i.e., shared knowledge of clinical experience amongst clinicians; 3) vicarious trauma encountered by clinicians; and 4) refugee status advocacy—e.g., the labeling of a precarious political status [18].

**Research gap.** A gap in knowledge exists about how integrated mental health care and services work for refugees who are a subgroup, of people who have experienced forced migration. To address this knowledge gap, we took a realist approach to understanding how complex systems work to promote refugee mental health and access to care in Canada. Realist research is theory driven; methods are used to evaluate complex social programmes and policies [29,46]. The research question guiding our work is, "How, when and why does integrated mental health care work in resettlement settings?"

We need to know more about how mental health systems and services work across different service types and as an integrated whole. Specifically, we need to identify the contexts in which these systems work well, and to determine what system-level conditions contribute to their success. Currently, most studies focus on program-level interventions at the micro level—i.e., mental health interventions within specific settings that provide mental health treatment such as psychotherapeutic interventions [15,16,18]. Moreover, primary health care is often referred to as the first point of contact for refugees [3], yet not all primary care settings are able to address social determinants of refugee mental health. In host countries that receive refugees, settlement services are often the first point of contact for refugees, providing a broad range of social and health care services such as language training, labor market preparation and access to health care services [22]. While research has provided insights into possible barriers and facilitators to accessing mental health services among refugee population groups, there are currently no systems in place that account for how mental health services work, in what contexts they work or for whom they work well.

**Study aims.** Our aims are to develop a program theory of integrated mental health care for refugees in resettlement settings in Canada, with the long-term aim of testing the theory against the global scientific literature. As realist approaches are theory-driven, we aim to explain how interventions work in real world contexts and to understand the

contexts that trigger actors' behavior or performances within refugee resettlement settings. Our third aim was to build upon existing theory on understanding the sociocultural and contextual forces underpinning practice delivery outcomes for diverse migrant groups. To this end, we adopted a realist approach for understanding the working mechanisms at play related to various actors' choices and actions that appear to cause positive or negative outcomes to happen [34,48].

## Materials and methods

### Ethics statement

Research ethics approval was obtained by the University of British Columbia and the University of Victoria, BC; REB Number: H22-03195. We used a harmonized review process because we interviewed interest group holders across jurisdictions and those who worked in multiple health authorities. Study recruitment occurred between 11/21/2023-02/16/2024. All participants provided informed written consent to take part in the deliberative dialogue sessions, to have their interviews audio recorded and transcribed for the purpose of this research.

### Study design and setting

We applied a realist methodology informed by community based participatory research (CBPR) principles to guide our process. CBPR is a flexible approach that must be adapted for diverse community partnerships [49]. Fitting with realist methods, we applied principles of CBPR into the design, data collection methods with interest group holders who participated in this research [50]. We adopt the MuSE consortium term 'interest group holders' to refer to community participants who have a legitimate interest in and responsibility for health-related decisions affecting refugee populations [51]. Drawing on CBPR principles we strived to first, include synergistic values of healthcare equity and a shared commitment to addressing refugee mental health with interest groups who participated in this research [50].

Second, we adopt cultural humility to our community partnership building [52]. This means that we are reflexive of our own researcher positionality and perspectives and how our values shape knowledge production [53]. NC (first author) identifies as a first-generation immigrant from Palestine and the former Yugoslavia, is a community mental health nurse and researcher but novice in realist methods. AAB (second author) is an early-career mental health researcher, a naturalised British citizen of Spanish heritage with dual nationality, and a scholar applying realist approaches to his research practice. MH (third author) originates from the US and identifies as a health researcher with growing roots in realist methods and social researcher with personal and professional motivations for addressing structural discrimination and health injustices. FCM (last author) originates from Cameroon and is an early career researcher with experience in applying realist methodology and approaches to evaluate mental health services and policies. As researchers we acknowledge our varying degrees of research expertise in applying realist methods from novice to expert but value the strength that our multiple perspectives bring to enhancing theory building with other interest groups. Importantly, we address power imbalances by respectfully including diverse ways of knowing with all interest groups involved in this study. Third, we view integrated care for refugees through an ecological and multidisciplinary perspective where integrated mental health operatives within broader community, societal and geopolitical forces [54]. Fourth, we strive to engage in equitable research partnerships by building trust, having multiple opportunities for dialogue and commitment to capacity building, albeit in resource constrained settings. In this research we continually engaged interest groups in four sequential knowledge translation activities including a) a realist workshop; b) deliberative dialogues; c) a World Café event and d) a consensus building exercise using Miro board (an online collaborative workspace) to brainstorm and share findings from our deliberative dialogues. Although, each activity provided diverse sources of knowledge to help inform our initial program theory (IPT), we only present findings from the deliberative dialogues. Our fifth CPBR principle was to promote interest group self-efficacy by sharing values and perspectives which has been shown to produce meaningful outcomes for community increase knowledge translation [50].

PLOS Mental Health

## Participant recruitment and description

We used purposive criterion-based sampling strategy, alongside snowball sampling techniques [55,56] to recruit diverse interest group holders who had expertise in providing mental health services to refugees and asylum seekers. We deliberatively included participants with lived experience of forced migration who accessed mental health services as they are typically not included in research that directly affects them [57,58]. Initial recruitment was done after holding a community-based realist workshop. Following the community realist workshop, interested interest groups were invited to contact (NC) for a follow up deliberative dialogue interview that was conducted in person or online and either with an individual or as a focus group. Snowball sampling occurred through NC's known networks, i.e., local immigrant and refugee health sector table. We combined a reflexive thematic analysis with realist theory building; as such sample size was not predetermined rather, we analysed our data in terms of informational power, theoretical saturation, and richness to address our research aims [59,60]. Participants included interest group holders and are representatives from a settlement service organization, a community health center, a mental health organization, a ministry of health policy analyst, a survivor advocacy group and a specialized immigrant and refugee community health clinic (N = 24). See Table 2 for a breakdown of the participants.

**Deliberative dialogue interviews.** Our interest group holders suggested the term 'deliberative dialogues' to describe conversations that promote evidence-informed policy [61]. Deliberative dialogues are purposeful conversations that build capacity for privileging tacit knowledge and real-world experiences of diverse practitioners used to inform and engage policy makers [61]. In this paper, we present an analysis of the findings from our deliberative dialogues, which took the form of individual interviews and a focus group (conducted online and in person) to understand what worked in resettlement contexts to promote integrated mental health care. We consider this stage of knowledge as what Graham et al., (2006) refer to as 'first generation' knowledge because it requires further refinement with second generation of knowledge, i.e., a realist review [62].

Our deliberative dialogue interviews lasted from sixty to ninety minutes. Consistent with realist approaches and purposive sampling strategies in qualitative research, we leveraged the perspectives of a range of interest group holders'

**Table 2. Participating organizations and roles.**

| Organization (Total N = 24) | Roles |
|---|---|
| Settlement Service Organization (N = 8) | (1) Immigrant and Refugee Service Manager<br>(1) Director of Services for Counselling<br>(1) Resettlement Assistant Program Manager<br>(1) Settlement Worker<br>(1) Settlement Worker in Schools<br>(1) Settlement Client Navigator<br>(1) Manager of Immigrant Integration and Research Planning<br>(1) Resettlement Assistant and Case Management Worker |
| Community Health Center (N = 6) | (1) Manager of Equity & Engagement<br>(5) Community Health Workers |
| Specialized Refugee Primary Care Clinic (N = 4) | (4) Nurse Practitioners |
| Mental health organization (N = 1) | (1) Case Worker in Mental Health and Substance Use |
| Ministry of Health (N = 1) | (1) Ministry of Mental Health and Addiction Policy Analyst |
| Survivor Advocacy Group (N = 4) | (4) Lived experience participant[a] |

[a]Some members reported having relevant lived experience as well as providing refugee resettlement support.

representatives to glean different expertise and perspectives [63–65]. Using a semi structured interview format our deliberative dialogues included a brief explanation of realist theory and example of a mechanism and the following guiding questions: 1) What do you think integrated care is? 2) what do you think promotes integrated mental health care for refugees; 3) what are the resources needed to provide integrated care? and 4) what is the mechanism thought to create integrated care? (see S1 Text).

Cognizant of the different power differentials between professionals and people with lived experience [65,66], we held the deliberative dialogue sessions separately, such that each dialogue session was held with representatives from the same interest group. This approach helped to facilitate trust between the participants and facilitator and enhanced the opportunity for participants to freely discuss their perspectives. All interviews were conducted online and in-person by the first author (NC). Audio recordings were transcribed verbatim and stored in the University of Victoria's hard drive system.

**Analysis.** Our analysis of the deliberative dialogue interviews included several iterative phases. The iterative approach in realist data analysis is justified by the realist commitment to ontological depth and fallible knowledge, recognizing that understanding reality requires moving back and forth between data and theory to refine explanations that have strong explanatory power and are both empirically adequate and ontologically plausible [33].

Before initiating the analytical phase, the team (NC, AAB, MH and FM) met twice virtually to read one of the interview transcripts together. This process ensured alignment in the realist analysis process among all team members. The initial interview transcripts were read by NC and coded independently by two researchers (AAB and MH) using the dyad and triad method [67]. This method involves examining dyads (two linked elements, such as context–mechanism or mechanism–outcome) and triads (full context–mechanism–outcome configurations, or CMOs). This approach enabled our team to identify relationships between the contexts, underlying mechanisms and causal outcomes.

We then adopted Wiltshire and Ronkainen's realist approach to thematic analysis to aid in pattern identification across the data [68]. Wiltshire and Ronkainen's realist thematic analysis [68] integrates realist philosophy with reflexive thematic analysis by organizing themes across three ontological levels: experiential (what participants said), inferential (what can be plausibly interpreted or generalized) and dispositional (underlying causal powers or mechanisms) (see S1 Table). Experiential themes were initially developed by AAB and MH by collectively reading and discussing the data to identify themes that closely represented participants' experiences. At this stage, quotations were extracted as evidence of the experiential theme, and dyads and triad configurations were used to explain the existence of the mechanisms and context relationships. After experiential themes and quotations were developed and extracted, the analysis moved beyond firsthand-level accounts to develop inferential themes (see S1 Table). Inferential themes go beyond what participants say to identify unobserved explanations and occurring experiences [68]. This approach allowed our team to think both "inductively to infer what might be occurring in broader contexts and abductively to redescribe the theme in more conceptually abstract terms" [68]. From here, AAB and MH developed linked dispositional themes or hypothetical causal explanations based on the relevant literature. NC read all the experiential, inferential and dispositional statements to validate or refute the emerging explanatory theory for each experiential, inferential and dispositional statement. During this collaborative data analysis process, we negotiated differences through discussions and several iterations of dispositional themes, e.g., NC would clarify context of participant statements. We were also reflexive of our positionality as each of us have varying degrees of expertise and experience in either the methods or expert knowledge. We acknowledge these positions not as a performative act but to describe how our positions influenced our analysis of the data [53]. This negotiation of difference is common when data analysis is conducted by researchers who did not conduct the interviews, and where one researcher has 'insider' knowledge and can enhance interpretive validity [68]. In our case, contextual and specialized knowledge about the resettlement landscape was shared by NC, who had conducted the deliberative dialogue interviews as well as the realist workshop and the world café with interest group holders.

The final stage of our analysis aimed to explain the underlying causal powers or mechanisms that must exist for observed patterns to occur (i.e., identify dispositional themes). This stage used retroductive reasoning, asking: What

must be true about the world for this pattern to make sense? AAB and MH worked independently to arrive at dispositional themes and then came together to develop a synergy table and explanatory statements, which are succinct interpretations that articulate how deep, often-unobservable mechanisms (dispositional themes) help to explain abstract patterns (inferential themes) which are grounded in participants lived experiences (experiential themes) (see S1 Table). At this stage, NC, AAB and MH met regularly to reach consensus on all the dispositional and explanatory statements. NC and FCM provided iterative feedback and refinements on all dispositional and explanatory statements through comments in Microsoft Word and verbally during twice-monthly meetings.

We then constructed statements based on the explanatory statements, again using retroductive theorizing, to develop propositions about how, why and for whom integration works and in what unique contexts [35]. Our statements followed the ICAMO ("Intervention–Context–Actor–Mechanism–Outcome") heuristic, which is an adapted version of the traditional CMO ("Context–Mechanism–Outcome") heuristic used in realist research. See Fig 1 for a model on the interaction between each ICAMO element. When constructing ICAMO explanatory statements, elements of the heuristic may be in any order, if the statement conveys how each element reacts to one another to create some outcome.

All ICAMO statements were compiled into a Word document by AAB, with each component ("I or C or A etc.…") color coded for clarity. This color coding helped us to distinguish between the different elements and facilitated abstraction at the level of the context and mechanism. We analyzed the ICAMOs for patterns, including overarching contexts for integrated mental health care and similarities and differences at the mechanism level. Broader explanatory principles were then developed, synthesising the key programme theories or mechanisms of change.

This process involved making a judgement about the explanatory ICAMOs and was guided by (NC)'s *in situ* knowledge and whether the themes are defensible based on frontline providers' experiences as well as what is known in the literature [68]. We used a consensus document (S2 Text) to determine the accuracy and validity of the dispositional statements and evolving ICAMOs. In the following, we present the findings under each of the four micro-theories followed by two ICAMOs with corresponding excerpts and brief description. Guided by a CBPR approach we conducted a consensus building activity to refute or validate our IPT and co-develop priorities for action with interest group participants. Through sharing our IPT most participants concurred with the enabling and hindering mechanisms thought to trigger behaviours which promote integrated mental health care. Participant perspectives from the consensus building exercise were further analysed with the theoretical literature as part of our discussion section. In our case we describe our findings as micro theories to highlight a realist informed analysis and to differentiate from constructivist approach to results reporting which typically group constructs as themes.

Each step of the analysis is represented visually in Fig 2, below. At the end of the following results section, we present an IPT diagram, which provides a conceptual model of the demi regularities or patterns of the data.

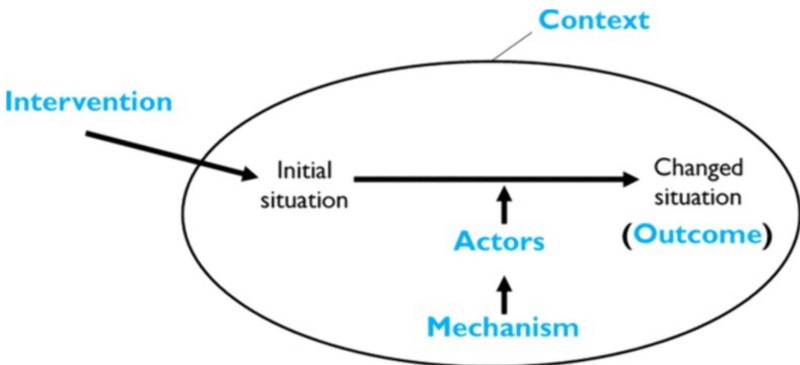

**Fig 1. ICAMO configuration model.**

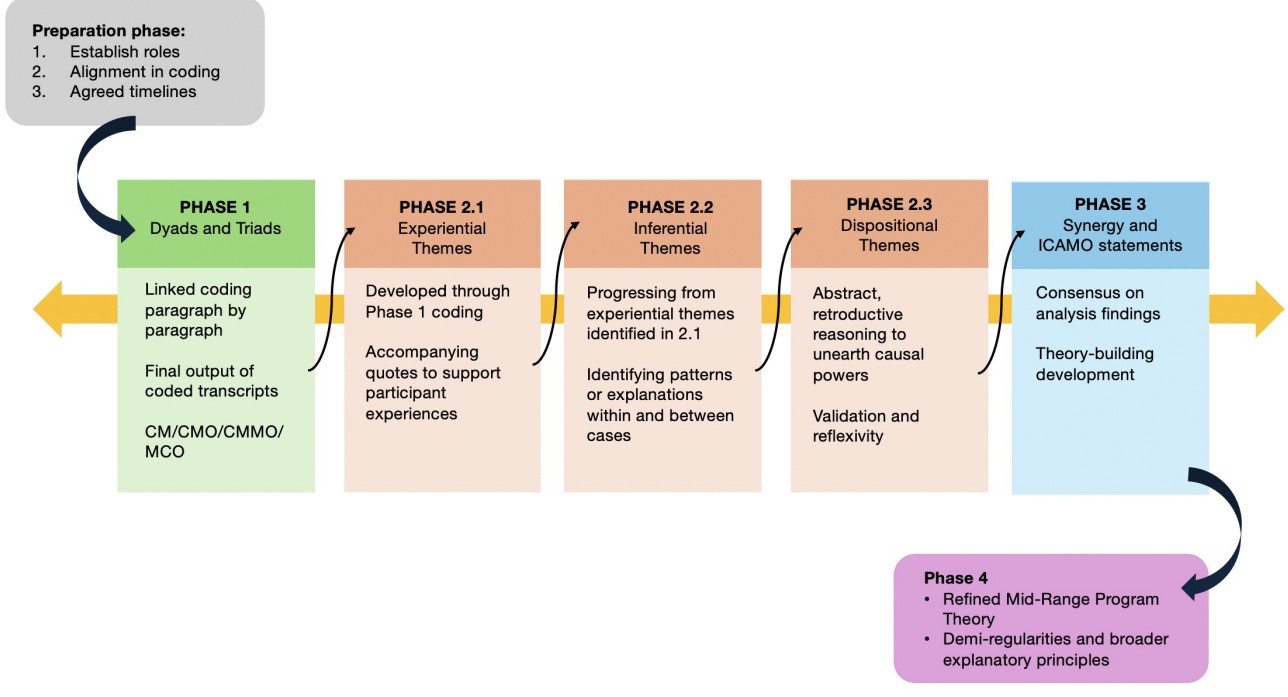

**Fig 2. Analysis process.**

## Results

Our findings suggest that integrated mental health care for refugees is improved when services are culturally and linguistically responsive; when system navigation is simplified and collaborative; when frontline providers are supported to deliver integrated, relational care; and when funding and policy environments incentivize long-term, holistic models rather than short-term crisis responses. Although these findings may be obvious to many who work within settlement and primary healthcare services promoting refugee mental health, our aims were to move beyond the surface level and to uncover the unobservable causal powers of how and why integrated mental health care works, for whom and in what contexts. In this regard, we wanted to show connections and relationships that are not directly observable [64].

We developed thirty-six ICAMOs and configurations (see S2 Text), We organized ICAMOs into four broad micro-theories: 1) cultural and relational factors, included 12 ICAMOs pertaining to how culture, language, stigma and social frameworks shape access and engagement; 2) system navigation and service access, included 7 ICAMOs related to how people move through mental health and social services; problems with silos, complexity, fragmentation; 3) organizational and workforce challenges: how agencies and providers operate internally, accounting for staffing, burnout, rigid workflows, reliance on champions or a nonclinical workforce, included 11 ICAMOs and, 4) policy, funding and governance barriers: funding, political will and structures that shape the whole system, included 6 ICAMOs. Through this clustering, we identified frequently occurring patterns or 'demi' regularities [50]. Below we included two ICAMOs per each micro theory with exemplary quotes to illustrate what enabled integrated care to work, in what contexts and why. We applied a realist phraseology to create explanatory statements [69]. For example, for each micro theory we applied the heuristic of 'IF/WHEN…THEN…BECAUSE' statements to represent realist theory building. IF/WHEN…THEN...BECAUSE correspond to context (C), outcome (O) and mechanism (M) respectively. Using the heuristic ICAMO we added (A) actors because they conduct behaviours that give rise to outcomes. Likewise, we added interventions (I) to differentiate from surrounding contexts that trigger mechanisms or resources thought to produce change.

## Micro-theory 1: Cultural and relational factors

**Alienation.** When refugee clients (A) access community health services without culturally tailored community-based care (I), they encounter mental health services that rely on Western, individualistic models that treat everyone the same—regardless of, for example, language ability (C), which can trigger feelings of alienation (M) for refugee clients. These feelings may lead to disengagement from services (O).

> *"We have cross-cultural health promoters, but if they are in a mental health crisis and we're trying to refer out, [but] we can't find a Spanish-speaking counsellor or a Dari-speaking counsellor. That's the problem. So even if we solve it on our end, there's nothing out there in the languages that we need." – Manager, Equity & Engagement, Multicultural Family and Health Center*

> *"Fitting scheduling those kids in the places and if the mother doesn't know how to drive and they're only depending on the father, maybe there's no father, you know, whatever the situation is—just, that's why, I think, many of them also stop because, like, I can't handle, you know, going and bringing and scheduling and finding, you know, that that timing. So when the school offers, you know, the counselling, they say, 'do whatever you want, but [only] if it's at school.' But going back to counselling at school, either they are overwhelmed with the cases they have in their own school, or they say, 'we're not equipped enough for handling the trauma and the issues.'" – Settlement worker in school*

In contexts where models of service delivery do not support the diversity of refugee clients' needs for, for instance, language interpreters or culturally tailored counselling support, the mechanism of alienation can result in client disengagement from services which in turn can increase barriers to mental health care.

**Trust.** When refugee clients (A) are introduced to culturally grounded peer support community mental health initiatives (C), which promote mental health awareness and engagement via peer and community figure presenters (I), they are likely to feel more connected to these services (O) due to their trust (M) in similar individuals with lived experience.

> *"That's important that your client feel connected to that. If they're not feeling connected so that doesn't work, and trust is important. Like, you have to know them; you have to know their culture. You have to respect and so then that they can trust first. And, later, you can provide that techniques and I think, in Canada, it happens a lot that the, the counseling is not culturally informed." – Lived experience participant*

> *"And with domestic violence, we've got a lot of families with domestic violence. It's a lot of work, especially if the woman decides 'I don't want to have this anymore.' And if she wants to leave with the children, then you have to find the transition house where she's understood having to settle her in and then the man also needs help." – Director of Services and Counselling for Immigrants and Refugees*

Participants discussed the importance of understanding cultural differences within and between diverse refugee communities and responding to them effectively. If interventions, such as support groups, are facilitated by lived experience, culturally responsive peers, clients may be more likely to trust services, leading to increased willingness to access and sustain their engagement with the care system.

## Micro-theory 2: Silo, complexity, fragmentation

**Stagnation.** When refugee clients (A) attempt to access community mental health support and services (C), they often encounter narrowly defined service mandates and lack of resource linkages (I) that block their ability to get the social care and healthcare they need (O), due to system failure/stagnation (M).

*"Yeah, so after they get discharged—yeah, unfortunately, some of them end up just using walking clinics or urgent care. Some find GPs and, to maybe a handful, we try to assist them in finding a GP, but we just need to do that because we need ongoing capacity, basically, because we get probably 20 referrals a week on average." – Nurse Practitioner*

The stalling and delay of the care routine, or stagnation, is a mechanism that hinders integration and access to health care resources. Stagnation happens in the context of narrowly defined organizational mandates, which often occur when specialized refugee clinics can only provide services for a short period of time, for instance, for only one year—and when attachment to primary care is difficult. Stagnation can result in clients turning to care services like urgent care that may not be capable of providing sustained or appropriate care. If these narrow mandates are operationalized in a fragmented service ecosystem, care becomes stagnated, leaving clients who need mental health care to take alternative routes to care—routes that often are unlinked to the mental health care system.

**Moral commitment.** When refugee clients (A) need urgent care (C), coordinated, bottom-up advocacy-driven approaches that address social determinants of health, such as housing or employment (I), result in better comprehensive care (O) due to service providers' moral commitment to their clients (M).

*"So, they, exactly—so they do the medical screening and Doctor [name], we agreed with the doctor there, even if they don't need, it he's referring everyone and he said, 'I want to do this for everyone.' Yeah and I think you're getting lots of referrals because of that." – Resettlement Assistant Program Manager*

Despite the barriers to integration of mental health care, providers are motivated by their ethical and moral commitment to ensure that refugees get the care they need. Moral commitment acts as a mechanism for increasing referrals and, potentially, equity-promoting mental health care supports to refugee clients.

*"Sometimes you need someone to help you to walk you through the resources. Because for someone who is new, that's going to take a lot of time to decide, 'OK, I have to make a phone call,' especially if there's a language barrier and all the different stages that you have to go through. I think, helping [it's better to be] them to get to resources instead of leaving them alone with resources." – Lived experience participant*

That providers are helping people access the resources they need suggests that a provider's moral commitment is a mechanism for change, as standard procedures do not often support refugee clients who need support beyond a certain threshold of responsibility. Resources like cultural brokers and community care workers who support social determinants of mental health needs for refugees are often motivated by compassion and moral commitment. These community workers enact an ethic of care and help refugees get to resources they need. Community workers are motivated because they themselves have often experienced hardship as refugees. If service providers working in bottom-up care procedures within urgent circumstances and complex care ecosystems foster a moral commitment to their work, then services across the system may be better prepared to receive and support refugee clients.

### Micro-theory 3: Organizational and workforce challenges

**Connection.** When refugee clients (A) are being served in a mental health environment with clinician shortages and limited-service capacity (C), the introduction of trained and supervised volunteers to provide community-based mental health support (I) clients are more apt to overcome access barriers (O) due to a sense of connection with the volunteers (M).

*"So, I think for cultural broker, it's more, it's a more direct support for the client. If there's a gap in between that [the] client cannot navigate it by themselves, the cultural broker would be a resource to help them to navigate that." – Lived experience participant*

*"I think the way this system is designed is not to keep people forever. And that's an active conversation right now, because relationality is a huge part of somebody with mental health challenges. They want to be connected. To most of the people, most of the people we 'discharge' in the quotation, often are back in the system within couple of months or weeks. And then the question, is it even necessary to discharge?" – Mental health provider*

Connection is a mechanism that promotes engagement and access to mental health resources in settlement contexts serving refugee communities. The conditions of mainstream care suggest that mental health services offered within a short time frame may prematurely disconnect people from services. Therefore, if trained and supervised, community representatives could be recruited to complement existing services for refugee clients, there is greater likelihood of improved access to services due to connections between clients and volunteers.

**Burnout.** When mental health providers (A) provide care in under-funded community mental health services with under-staffing and high turnover (C), the inability to provide integrated care services for clients (I) results in inconsistent client engagement (O), due to provider burnout (M).

*"They need to be funding for, for the integration for every team and, and getting, and staffing right, right? Right now, everybody—like, all the involved agencies—we, you know, social agencies and the health authorities are willing to come together and provide funding, resources for that, but at some point, they will be need to, like, specifically like as, as integrated services for mental health and substance use are expanding. They will be need for more training, more money, more people." – Ministry of Mental Health and Addiction Policy Analyst*

*"There's a lot of burnout in our wider profession. You know, there's new people that you have to rebuild their relationship with all the time and different service providers, and it takes time too and programs also change. So that's one of the challenges that we see." – Community Health Worker*

Burnout is a mechanism experienced by providers. It hinders integration of care by causing staff turnover, which leads to service disruption for refugee clients who have established trust with their provider and services. From a policy perspective, burnout could be mitigated by increasing funding for human resources and for training and support for mental health and addiction services. When mental health services are already facing human resource constraints and underfunded, service fragmentation is exacerbated. This fragmentation then adds pressure on service providers, leading to more burnout and ultimately fractured service delivery.

### Micro-theory 4: Policy, funding and governance barriers

**Fragmentation.** When refugee clients (A) have different immigration status (e.g., public versus private sponsorship) (C), differential access to resources (I) can result in inequitable services for clients (O) due to fragmentation of complex care needs (M).

*"We used to have more government-assisted refugees as a partner clinic, but with the dissolution of bridge clinic, that that work being privatized as well as the federal government now increasing privately sponsored numbers versus government assisted, hence kind of offloading some of the responsibility to community. We're seeing more privately sponsored [refugees] and I think that does affect how people settle, because they simply don't have the same level of support that government assisted refugees have, right?" – Nurse Practitioner*

When refugee status determines access to health care resources, we see inequitable differences in outcomes. For example, privately sponsored refugees often receive fewer, less substantial resources than refugees receiving government support. These stipulations force organizations to deliver care differently due to service fragmentation. Fragmentation

  

therefore hinders holistic, equitable and integrated mental health for refugees and other people with forced migration backgrounds, including those without refugee status.

**Proactivity.** When funders and health system leaders (A) in settlement settings (C) adopt a strategic focus on prevention and mental health promotion (I), this strategic focus can lead to efficient, cost-effective and client-centered intervention (O), because providers are proactive and share information (M).

*"So, kind of the I, […] think for the integrated teams, it needs to be—there is a need for information sharing between different agencies and, and different types of specialists within the team. There is a need for, kind of, team integration so there is a psychiatrist and a nurse and a social worker that are kind of physically collocated. And then can work with clients as a team. Both sharing data among themselves and then coordinating their efforts and being able to come together, talk to client as a team and then take cases back to their kind of respective agencies." – Ministry of Mental Health and Addictions Policy Analyst*

Settlement organizations are generally proactive when sharing resources, even when resources are scarce. This proactivity is a mechanism that enables the sharing of information, and therefore better integration of services and mental health supports, provider feedback and the sharing of information and knowledge helps to improve cohesive service provision.

These ICAMO statements are intertwined, impacting one another. Their connections are depicted in Fig 3.

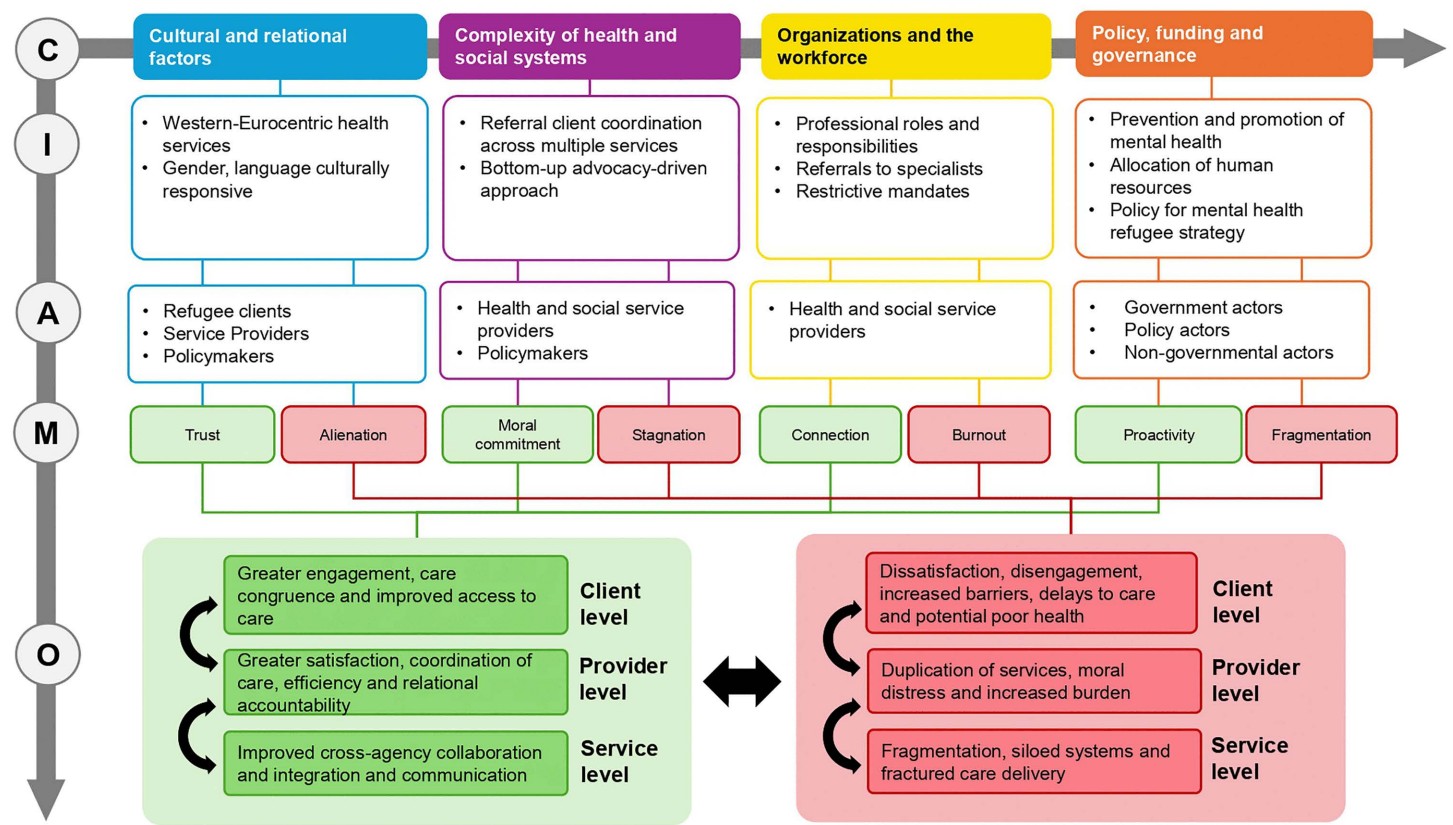

**Fig 3. An initial program theory (IPT) diagram.**

## Discussion

The aim of this study was to develop an IPT through deliberative dialogues in which we captured interest group holders' perspectives on what they perceived to promote integrated mental health services for people with forced migration backgrounds. We adopted a realist theory-building approach by constructing ICAMOs to identify and explain which interventions or behaviors are thought to promote integrated care for refugees, as well as which contexts and underlying mechanisms influenced the outcomes (positive or negative).

### Client level

Our micro theories (cultural and relational factors, system navigation and service access, organizational workforce challenges and policy and governance) resonate with previous realist studies conducted in contexts such as COVID-19 and mental health crises [43–45]. However, to our knowledge, program architecture for refugee mental health services has not yet been explored with depth. We concur with previous studies that call for mental health services that are adaptive and responsive to needs of refugees [12–16] and our findings suggest that trust is a consistent mechanism that enables integration of care across client, providers and services. At the client level, clients who have migration histories may experience distrust of institutional and government systems based on their ethnic, political or gender affiliation [70]. In our study, trust also resonated with settlement and health care providers in terms of enhanced service collaboration with diverse actors including counsellors and primary health care providers. Having a history of good working relationships also helped facilitate trust amongst providers. We found that primary health care services increased trust with refugee clients through use of cross-cultural health brokers and settlement workers in schools. Trust enhanced collaboration and promoted engagement and access to mental health resources for both children and parents from refugee backgrounds. This finding supports the existing evidence, as studies suggest that, when service organizations draw on each other's experience and expertise, decision-making becomes collaborative and promotes trust, and mutual respect in addition to making services resilient and responsive [45,46]. Similarly, trust can be enhanced to understand different contributions of partners from different agencies working under pressure [71].

Alternatively, alienation hindered client engagement with services, triggered by service conditions that are misaligned with client needs. Our findings suggest that mental health services must adapt to develop and provide equitable and culturally tailored care. This finding is well-known amongst clinicians; however, in the context of our study, we link the importance of systems-level and cross-sectoral involvement to the provision of this need. This linkage includes a multisectoral approach that extends beyond the settlement sector and includes the primary health care sector and formal mental health services. Given the complexity of refugee mental health needs, settlement sectors tend to carry the burden of responsibility for refugee mental health. The dominance of settlement organizations in supporting refugee mental health may constrain integration of cross-sector collaboration.

In addition, contexts that are related to alienation include Western, Eurocentric approaches that may not resonate with diverse identities and cultural backgrounds of refugees [11,12,15]. Increasingly, settlement organizations are adopting an intersectional perspective to better understand the overlapping dimensions of health for vulnerable population groups [72,73]. If researchers and policymakers were to adopt this critical theoretical approach, our findings could better enable clinicians to understand refugee mental health needs [73]. A critical rapid review on mental health of immigrants highlights that interventions that foreground context, culture community, relationality and power were found to be important mechanisms for promoting mental health support [15]. Based on interest group holders' perspectives in our research, resources such as cross-cultural health brokers, settlement workers in schools, community health workers and client navigators facilitate trust and mitigate alienation for refugee families. The relational nature of sociocultural and contextual forces underpins service delivery and consequent outcomes that are intended and unintended [15]. Findings from our consensus building exercise added that at the client level, service consistency, remain important contexts for promoting trust which ultimately resulted in improved health literacy, system navigation.

## Provider level

In many cases, such as in the survivor's advocacy group, service providers are also members of refugee communities and provide important cultural knowledge, language support and understanding of social determinants for refugee mental health. Lack of knowledge and cultural knowledge creates alienation and potential disengagement from services [12,15,18].

We found that many services were providing care in resource-limited settings due to funding constraints. However, the moral commitment of providers within settlement sectors and specialized primary health care clinics was evident in their upstream advocacy and expanded mandates to accommodate mental health service access. Service provider moral commitment was also evident as they extended their service to meet the social determinants of mental health care for refugees, i.e., including supports for health system navigation, language and cultural resources, and housing and counseling supports. These findings resonate with previous research in mental health crisis contexts in which compassionate care was a key mechanism related to health and service system values and leadership crisis [47]. Our findings suggest that moral commitment is enacted in contexts of resettlement, which in turn suggests that an underlying ethic of care and values of equity promote integration of mental health services for refugees. However, health and social service providers are bound by social and political conditions that affect policy while adhering to ethical guidelines that frame their practice [4,14]. During our consensus building exercise, interest group holders concurred that moral commitment and advocacy are important mechanisms for enabling greater satisfaction, coordination of care, accountability, while also stressing that emotional safety for providers and mitigating burnout required more resources. Although moral commitment is laudable it may not be sustainable in the context of governmental cutbacks, ultimately leading to a less resilient and integrated healthcare system.

## Service level

Service coordination and collaboration are hallmarks of integrated care [23]. Our study found that, in context of silos between services, notably between settlement and primary health care, stagnation occurs because of lack of collaboration. Stagnation is theorized as a mechanism that hinders access to care and results in system fragmentation. Ultimately, stagnation leads to delays in care for refugees who require interventions that promote their mental health. When services operate in silos and when no clear policy is in place for refugee mental health standards of care, such gaps can lead to refugee clients being referred to many different agencies without reaching an intervention because agencies are working in isolation—and, in some cases, are even competing against each other. This finding is consistent with mental health crisis contexts in which there is no developed response from multisector agencies [47]. Stagnation also happens when service mandates are restricted and informed by immigration government policy which structures whether forced migrants receive health care access because of their mode of entry—that is, because they arrived through government rather than private sponsorship [14]. Increasingly, immigration settlement sectors have experienced reduced government funding because of broader political forces, including racism and xenophobia [4,14,17,22]. As noted by interest group holders in our study, many health and settlement services providers are bound by policies that inform which refugees they can see and for how long they can see them when they offer mental health support through community-based referrals.

Our research further suggests that burnout is a mechanism that occurs in contexts in which providers are overburdened with their workload, and is a complexity associated with provision of refugee mental health. Ultimately, service provider burnout leads to decreased human resources and can significantly disrupt integrated care and relational continuity with refugee clients. Burnout is a mechanism that may also result in provider turnover when they are not able to provide services; as one provider noted, "…We need ongoing capacity basically because we get probably 20 referrals a week on average." Although our study did not focus on trauma services, it is well known that many refugees arrive with trauma histories that may be exacerbated in post-migration contexts [2,7,18]. This added complexity, combined with demanding workloads and limited resources, are contexts that can result in lower services and service provider burnout. We concur

with other research [18,47] that argues that understaffing and limited human resources may add additional barriers for accessing mental health support, as clients may have to retell their trauma histories multiple times to service providers at different agencies, potentially re-traumatizing them. These findings resonated with our interest group holders. During the consensus building exercise many participants acknowledged that complexity of refugee client's needs required broader systemic support. Participants drew attention to how political and social factors were stigmatizing for refugees which overlap with the need to develop trust, increased advocacy and moral commitment.

Our findings concur with other realist studies that advocate for more participatory, collaborative strategies across systems and services that engage in task sharing and task shifting [46,47]. While our research did not emphasize mechanisms of task sharing, many providers and people with lived experience described connection as a mechanism across client, provider and services. Connection enables providers and services to share knowledge and resources about their refugee clients [16]. Connection can also strengthen relationships across services through other human resources such as nonclinical staff, cross-cultural brokers or health navigators. For example, researchers found that when professionals train nonclinical service providers from the same background as the targeted client population (e.g., language, culture, migrant or refugee experience), these nonclinical staff can address mental health service gaps while empowering clients who are members of their own community [15,74]. However, nonclinical support for refugees is often in context of precarious resourcing [15].

Research in the context of COVID-19 found that social connectedness was an important mechanism for motivating governments to provide extra funding to mitigate social isolation so that members of vulnerable communities could access mental health resources [45]. Similarly, interventions such as social prescribing enacted by community workers provided cost-effective measures to integrate mental health resources [46]. These findings resonate with promoting integrated mental health for refugees, as settlement agencies now regularly connect with primary health care services, settlement supports and mental health resources through nonclinical providers such as cross-cultural brokers and community health workers, in constrained resources settings with limited funding.

We heard that lack of government funding contributes to constrained resources required to promote better integration across services and sectors. This lack of funding can result in service fragmentation, and an inability to innovate in resource development, such as the hiring and training of nonclinical support staff who can provide cross-cultural brokering. This suggest that lack of funding results in stagnation and fragmentation, leading to delays in care. In addition, a refugees may be referred to several agencies, i.e., 'revolving door' or cycling through agencies without reaching an appropriate intervention because agencies are working in isolation and with limited resources.

Integrated governance and partnership between health and social care services can facilitate support for sustainable financing [46]. A recent study conducted with refugees identified that physician and health care provider shortages are a top research priority to address service gaps in the Canadian health care system [22]. System fragmentation within mental health services may reflect this systemic structural gap. We suggest that systemic gaps could be mitigated by implementing connections with government officials and policymakers as well as sharing across services resources, knowledge and integrated budgets [46]. Connection in this context may facilitate adaptation and a more resilient health care systems for refugees. In the current governmental climate of fiscal restraint, many not-for-profit settlement agencies have experienced significant defunding to services [17], leading to a greater reliance on other organizations.

Integrated mental health care for refugees requires a different program architecture, that provides sustained funding for including migrant-friendly environments, culturally safe services and supports that prioritize social determinants of refugee mental health. Our findings resonate with studies conducted in crisis conditions like COVID-19, where the design and delivery of mental health services moved beyond the status quo to adapt to community-driven needs [45]. Building on previous theory-driven models of services affecting refugee mental health [16,59] our IPT suggests that mechanisms of trust, moral commitment, connection and proactivity work to support refugee mental health across client, provider and services.

These mechanisms could decrease the burden of responsibility of care in the settlement and primary health care services, which currently provide most mental health services.

**Potential candidate theories.** A core component of realist theory building is to apply candidate theories to serve as a starting point to guide evidence collection and analysis by proposing potential mechanisms, contexts, and outcomes of an intervention [33,38]. The theory of health equity in the context of health care ethics [75–78] may add value to theory-informed models of service and to program interventions aimed at promoting refugee mental health. Adopting health equity theories would mean that the distribution of health resources must consider the diverse contexts, backgrounds and intersecting determinants of refugee mental health [75,71]. This ethical objective is not to distribute care through status quo approaches; rather, to treat people according to unique needs, including providing access to language interpreters or to culturally tailored and gender responsive care [12,15,75]. Ethical standards of health equity stem from principles of justice in which access is prioritized for those facing the most or highest barriers and ensures that corresponding resources are in place that promote health [77]. Collectively, interest group holders in this study prioritized mental health through their advocacy and values of healthcare equity.

The WHO (2022) World Mental Health report argues that it is time for change, emphasizing the need for accessible community-based mental health services [20] in which care is closely integrated across multiple sectors to address the full range of needs of people with mental health challenges. Furthermore, adoption of human rights perspective on mental health service delivery and universal health care is needed; adopting universal and comprehensive primary healthcare means that access to mental health services must also be affordable and accessible and must include cultural awareness of the social and cultural factors that can facilitate refugee mental health [75,79].

Intersectionality theory posits that overlapping identities (gender, immigration status, language ability, and so on) can advantage or disadvantage population groups [80–83]. Status quo approaches may further disadvantage refugee groups and who may experience marginalization across multiple levels of care. Intersectionality is rooted in critical theoretical paradigms of knowledge [83] and could be applied to understand mental health inequities for refugees accessing mental health services. Also, intersectionality can be used to investigate the factors shaping inequities across broader health determinants and their interactions through macro (global and national), meso (regional and provincial), and micro (community and individual) levels [82]. At the crux of intersectionality theory is mitigating social and structural inequities by addressing systemic power relations [83]. This approach may shed more light on mechanisms at play that promote or hinder integrated mental health care for refugees across healthcare systems.

## Study strengths and limitations

Most research exploring underlying mechanisms of change have used rapid reviews, summarizing themes of mechanisms and theories of change. While such an approach is useful, our research drew from multidisciplinary interest group holder perspectives, including a survivor advocacy group representing refugee lived experiences, to glean their perspectives on what worked to promote refugee mental health. We also privileged the voice of settlement agencies and primary health care services who provide most of the mental health support to refugees in Canada. We adopted a three-step analysis process that 1) used the dyad/triad method by Jackson and Kolla to develop initial context mechanism and outcome configurations; 2) adapted Wiltshire and Ronkainen's realist thematic analysis; and 3) drew on Mukumbang et al.'s ICAMO hypothesis to refine our explanatory dispositional statements and thus to examine relationships between context, intervention, actors, mechanisms and outcomes. Although we found mechanisms that enabled integration of mental health services, this process also allowed us to visualize mechanisms and contexts that hindered integration of mental health services for refugees. This process promoted the rigor of our findings.

Because we did not concurrently conduct a realist review of the literature, we are unable to provide a mid-range theory. However, through our analytic process, theoretical validity was enhanced by our findings' alignment with literature that focuses on designing mental health services for refugees. Our IPT requires further validation and testing against a robust

review of the literature and ongoing participation of our interest groups [50]. We re-engaged with interest group holders in a consensus building exercise to share our analysis, engage in further dialogue and prioritize action. However, methodologists have argued that member checking may not be the most effective form of validation because viewpoints are not static and critical feedback requires ongoing dialogue and engagement [59]. We plan to include more opportunities for interest group holder engagement throughout our knowledge translation activities including a realist review to substantiate our theory building. Moving forward this research can serve as a building block to understand the broader conditions and mechanisms that shape integrated mental health care for refugees. Our study is also limited by lack of representation from government and policymakers and will be a key consideration for future research.

## Recommendations

Our findings show that mental health services for refugees are underfunded and operating in resource-constrained contexts. We recommend that government and policy actors provide funding for expanded nonclinical roles, including cross-cultural health brokers, settlement workers in schools and health system navigators, to support access to mental health resources and to promote the long-term mental health of refugees. Enhancing resources can facilitate trust, a mechanism shown to have system change effectiveness for marginalized population groups including refugees [16,46]. Investing in human resources and staff training can also promote provider knowledge and development so that moral commitment and ethical care practices are integrated across all services including mainstream mental health and primary care settings. Proactivity, advocacy and connections between government and policy and settlement services should also be prioritized to enhance sociocultural and contextual understanding of refugee mental health and to decrease burden of responsibility for care in the settlement sector. Currently, insights from policymakers are lacking, and establishing formal connections could mitigate barriers related to precarious resourcing. Future research should include evaluation of mental health services inclusive of settlement services and primary healthcare to address broader system gaps and integration. This omission has resulted in a sustained Eurocentric and individualist culture of care that has not addressed the complexity or social determinants for refugee mental health. Intersectionality theory and theories of health care ethics and equity may provide useful frameworks for advancing a theory of change and provide more understanding about what promotes integrated mental health care for refugees.

## Conclusion

Mental health research remains a top priority for people with forced migration backgrounds [23]. Our IPT suggests that trust between service providers and clients can facilitate engagement and culturally safe and supportive mental health services. Likewise, a morally committed workforce increases values of equity, fairness and just practices, which in turn can result in better health outcomes and improved access to needed resources. Building better connections across frontline practices and services, as well as with policymakers and government actors, can promote the prioritization of refugee mental health. Building such connections requires proactivity, which can improve interagency collaboration, communication and efficiency. However, our IPT is underpinned by contexts in which cultural and relational factors are undervalued, and in which complexity of care needs remain a priority as more refugees are forced to migrate due to war, violence and environmental change. Research suggests that crisis conditions force system change and adaptation [46,48] however, culturally safe and equitable mental health services and supports for refugees ought to be mainstream for interventions to work [20]. We invoke proactive research to move from theory to action and integrated knowledge mobilization to promote refugee mental health. This involves strengthening mental health services and mainstreaming intersecting determinants of refugee mental health.

## Supporting information

**S1 Text. Interview guide.**
(DOCX)

**S2 Text. Consensus-building exercise.**
(DOCX)

**S1 Table. Initial program theory synergy tables.**
(PDF)

## Acknowledgments

We wish to acknowledge our interest group holders for sharing their knowledge and time. We sincerely thank Susan F. Jackson for consulting on our context-mechanism-outcome configurations. Simon Caroll and Maura Macphee for reading a previous version of the article. We would like to thank Letitia Henville for editing an earlier version of this article.

## Author contributions

**Conceptualization:** Nancy Clark, Alejandro Argüelles Bullón, Mita Huq, Ferdinand C. Mukumbang.

**Data curation:** Nancy Clark.

**Formal analysis:** Nancy Clark, Alejandro Argüelles Bullón, Mita Huq, Ferdinand C. Mukumbang.

**Funding acquisition:** Nancy Clark.

**Investigation:** Nancy Clark.

**Methodology:** Nancy Clark, Alejandro Argüelles Bullón, Mita Huq, Ferdinand C. Mukumbang.

**Project administration:** Nancy Clark, Mita Huq.

**Resources:** Nancy Clark.

**Supervision:** Ferdinand C. Mukumbang.

**Validation:** Nancy Clark, Alejandro Argüelles Bullón, Mita Huq, Ferdinand C. Mukumbang.

**Visualization:** Nancy Clark, Mita Huq.

**Writing – original draft:** Nancy Clark, Alejandro Argüelles Bullón.

**Writing – review & editing:** Nancy Clark, Alejandro Argüelles Bullón, Mita Huq, Ferdinand C. Mukumbang.

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
