## [Decision Letter · Decision Letter 0]

15 Sep 2025

PMEN-D-25-00314

Integrated Mental Health for Refugees: A Realist Theory Building Study

PLOS Mental Health

Dear Dr. Clark,

Thank you for submitting your manuscript to PLOS Mental Health. After careful consideration, we feel that it has merit but does not fully meet PLOS Mental Health’s publication criteria as it currently stands. Therefore, we invite you to submit a revised version of the manuscript that addresses the points raised during the review process.

Your manuscript has been assessed by one reviewer, and their comments are available below.

Please note that we have only been able to secure a single reviewer to assess your manuscript. We are issuing a decision on your manuscript at this point to prevent further delays in the evaluation of your manuscript. Please be aware that the editor who handles your revised manuscript might find it necessary to invite additional reviewers to assess this work once the revised manuscript is submitted. However, we will aim to proceed on the basis of this single review if possible.

We look forward to receiving your revised manuscript.

Kind regards,

Jenna Scaramanga

Staff Editor

PLOS Mental Health

Journal Requirements:

1. Please provide additional details regarding participant consent. In the ethics statement in the Methods and online submission information, please ensure that you have specified (1) whether consent was informed and (2) what type you obtained (for instance, written or verbal, and if verbal, how it was documented and witnessed). If your study included minors, state whether you obtained consent from parents or guardians. If the need for consent was waived by the ethics committee, please include this information.

Additional Editor Comments (if provided):

Reviewers' comments:

Reviewer's Responses to Questions

**Comments to the Author**

1. Does this manuscript meet PLOS Mental Health’s publication criteria?

Reviewer #1: Partly

2. Has the statistical analysis been performed appropriately and rigorously?

Reviewer #1: Yes

3. Have the authors made all data underlying the findings in their manuscript fully available (please refer to the Data Availability Statement at the start of the manuscript PDF file)?

Reviewer #1: No

4. Is the manuscript presented in an intelligible fashion and written in standard English?

Reviewer #1: Yes

Reviewer #1: Thank you for the opportunity to review “Integrated Mental Health for Refugees: A Realist Theory Building Study” for possible publication in PLOS Mental Health. This is a relevant paper that explores the development of an initial program theory on the integration of refugee mental health across services. Research on this topic holds the potential to improve our understanding of how to better support people with forced migration backgrounds. However, some recommendations are offered to be addressed before this paper can be accepted for publication. I hope the comments are useful to you in improving the manuscript for publication.

First, the manuscript provides a rather superficial coverage of the relevant literature. The section could be improved by providing an overall description of the refugee and asylum seekers characteristics.

The introduction section could also benefit from specific examples regarding some of the information presented. For example, a detailed description of cultural stigma along with examples.

Similarly, the authors should describe interventions that have shown to work with this population.

For a better reading, please carefully review the organization of the introduction section avoiding unnecessary repetitiveness to improve its flow.

The authors state this study implemented a participatory research approach but provide very detail about it. If this study was designed as such, the procedure should be described in detail.

More description of the recruitment process and justification of such is necessary. A strong justification for the sample size is missing. The eligibility criteria is not explicitly stated, please provide this information and justification for such.

Also, the authors do not provide sufficient description of the characteristics of the participants.

Importantly, the format of the dialogue sessions is very vaguely described, leaving the reader with a very unclear picture of how these developed.

I appreciate the authors detailed description and justification of their qualitative analysis. However they do not provide the authors’ positionality which would enrich the paper, for example describing how their positionalities impacted the group experience and research overall following suggestions outlined by Grzanka & Moradi (2021) re: positionality statements:

Grzanka, P. R., & Moradi, B. (2021). The qualitative imagination in counseling psychology: Enhancing methodological rigor across methods. Journal of Counseling Psychology, 68(3), 247

The authors provide quotes from participants, but I suggest adding information about the participant such as their gender and age.

Some of the qualitative results are very repetitive, the authors could instead use the space to expand of their presentation of the findings.

There seems to be a lot of overlap in the results and discussion sections, the discussion section reads a bit more like a recapitulation of the results section. Please revise to be less descriptive of results and more synthesized with the extant literature in line with this goal.

The authors do a good job articulating the limitations present in the study.

**Do you want your identity to be public for this peer review?** For information about this choice, including consent withdrawal, please see our Privacy Policy

Reviewer #1: No

---

## [Decision Letter · Decision Letter 1]

19 Dec 2025

PMEN-D-25-00314R1

Integrated Mental Health for Refugees: A Realist Theory Building Study

PLOS Mental Health

Dear Dr. Clark,

Thank you for submitting your manuscript to PLOS Mental Health. After careful consideration, we feel that it has merit but does not fully meet PLOS Mental Health’s publication criteria as it currently stands. Therefore, we invite you to submit a revised version of the manuscript that addresses the points raised during the review process.

Please see below comments from Reviewer 1, which require attention.

We look forward to receiving your revised manuscript.

Kind regards,

Jason Cameron McIntyre, PhD; B App Sc; B Psy Sc (Hons I)

Academic Editor

PLOS Mental Health

Journal Requirements:

Reviewers' comments:

Reviewer's Responses to Questions

**Comments to the Author**

Reviewer #1:

Reviewer #2: All comments have been addressed

publication criteria?

Reviewer #1: Yes

Reviewer #2: Yes

3. Has the statistical analysis been performed appropriately and rigorously?

Reviewer #1: N/A

Reviewer #2: Yes

4. Have the authors made all data underlying the findings in their manuscript fully available (please refer to the Data Availability Statement at the start of the manuscript PDF file)?

Reviewer #1: (No Response)

Reviewer #2: Yes

5. Is the manuscript presented in an intelligible fashion and written in standard English?

Reviewer #1: Yes

Reviewer #2: Yes

Reviewer #1: Overall, this manuscript has improved notably and represents a valuable contribution to the literature. The addition of clear definitions of asylum seekers, refugees, and stigma, along with the inclusion of relevant additional literature, strengthens the Introduction. The section now reads more smoothly and provides a clearer conceptual foundation for the study.

However, the added sentence now makes the following section somewhat repetitive:

“The lack of integration of care across systems and services can lead to increased healthcare costs and poor mental health for forced migrants [14]. These challenges can result in inequitable access to and cost of mental health care and poor mental health outcomes [6].”

The authors did a good job addressing my previous comments regarding the participatory research approach. The added detail in the Study Design and Setting section improves transparency. Similarly, the inclusion of a brief description of researcher positionality in this section is appreciated and enhances reflexivity. However, the positionality of the second author “AAB (second author) is …” is currently missing and should be included for completeness and consistency.

Importantly, the format of the dialogue sessions remains very vaguely described. The authors state that “dialogue sessions were semi structured interviews” but it would be important to know what kinds of questions were included in such interviews.

The Analysis section is thorough, but it could be more succinct to ensure the manuscript remains within the journal’s length specifications. Additionally, the statement, “We applied a realist phraseology to create explanatory statements [71],” would benefit from brief clarification to support reader understanding, particularly for those less familiar with realist methodology.

Finally, in the presentation of themes, the “M” (mechanism) component appears to be missing for several themes, specifically stagnation, connection, and burnout. Addressing this will improve clarity and consistency.

With these minor revisions, the manuscript will be further strengthened.

Reviewer #2: All queries are addressed. The manuscript is scientifically sound.

**Do you want your identity to be public for this peer review?** For information about this choice, including consent withdrawal, please see our Privacy Policy

Reviewer #1: No

Reviewer #2: No

---

## [Editor Report · Decision Letter 2]

12 Jan 2026

Integrated Mental Health for Refugees: A Realist Theory Building Study

PMEN-D-25-00314R2

Dear Prof Clark,

We are pleased to inform you that your manuscript 'Integrated Mental Health for Refugees: A Realist Theory Building Study' has been provisionally accepted for publication in PLOS Mental Health.

Best regards,

Jason Cameron McIntyre, PhD; B App Sc; B Psy Sc (Hons I)

Academic Editor

PLOS Mental Health